# Mice Condition Cephalic-Phase Insulin Release to Flavors Associated with Postoral Actions of Concentrated Glucose

**DOI:** 10.3390/nu16142250

**Published:** 2024-07-12

**Authors:** John I. Glendinning, Alix Archambeau, Lillian R. Brouwer, Alyson Dennis, Kiriaki Georgiou, Jessica Ivanov, Rochelle Vayntrub, Anthony Sclafani

**Affiliations:** 1Department of Biology, Barnard College, Columbia University, New York, NY 10027, USA; lillianrbrouwer@gmail.com (L.R.B.); alysondennis96@gmail.com (A.D.); kirageorgiou@gmail.com (K.G.); jessicaivanov1a@gmail.com (J.I.); 2Department of Neuroscience & Behavior, Barnard College, Columbia University, New York, NY 10027, USA; asiarchambeau@gmail.com (A.A.); rvayntrub4@gmail.com (R.V.); 3Department of Psychology, Brooklyn College of City University of New York, Brooklyn, NY 11210, USA; asclafani@gc.cuny.edu

**Keywords:** classical conditioning, saccharin, glucose, maltodextrins, cephalic-phase insulin release

## Abstract

Rats can condition cephalic-phase insulin responses (CPIRs) to specific sounds or times of the day that predict food availability. The present study asked whether mice can condition a CPIR to the flavor of sapid solutions that produce postoral glucose stimulation. To this end, we subjected C57BL/6 mice to one of six experimental protocols. We varied both the duration of the five training sessions (i.e., 23 h or 1 h) and the nature of the training solution. In Experiment 1, consumption of a 0.61% saccharin solution was paired with IG co-infusion of a 16% glucose solution. In Experiments 2–6, the mice consumed a training solution containing a mixture of 0.61% saccharin + 16% glucose, 32% sucrose, 32% maltodextrin, flavored 32% maltodextrin, or 16% maltodextrin. We subsequently asked whether consumption of any of these fluids conditioned a CPIR to a test solution that produced a similar flavor, but which did not elicit a CPIR in naïve mice. The mice did condition a CPIR, but only to the solutions containing 32% maltodextrin. We attribute this conditioning to postoral actions of the concentrated maltodextrin solutions.

## 1. Introduction

When blood glucose levels rise above baseline during a meal, pancreatic β-cells respond by increasing insulin secretion into the bloodstream. Insulin promotes the transport of glucose into liver, muscle, and fat cells and thus helps return blood glucose levels to normal [1]. Insulin secretion can be augmented by cephalic (i.e., head) sensory inputs from foods [2,3,4], which trigger the release of acetylcholine (ACh) from postganglionic parasympathetic neurons in pancreatic islets [5,6]. When ACh binds to muscarinic ACh receptors in β-cells [1,7], it potentiates glucose-induced insulin secretion [8,9], causing insulin to be secreted more quickly and in greater quantities [2,10]. This cephalic-phase insulin response (CPIR) has a large impact on glucose homeostasis. Indeed, many different species of mammals exhibit impaired glucose tolerance when CPIR is not expressed—e.g., humans [11,12,13], macaques [14], rats [5,15,16,17,18], and mice [19]. There is also evidence for large individual differences in CPIR magnitude in mice and humans, and that subjects who exhibit relatively large CPIR magnitudes also exhibit better glucose tolerance [20,21].

At the turn of the 20th century, Ivan Pavlov demonstrated that when an arbitrary sound is repeatedly associated with the arrival of food, hungry dogs conditioned a salivation response to that sound [22]. Here, we tested the hypothesis that mice can condition a CPIR to the flavor of sapid solutions that produce postoral glucose stimulation. This hypothesis is topical for several reasons. There are concerns that regular consumption of foods and beverages containing low-calorie sweeteners (LCSs), sugars, and/or maltodextrins may alter insulin secretion and glucose homeostasis through a conditioning process (e.g., Refs. [23,24]). Second, there are marked individual differences in CPIR magnitude in humans [20] and mice [21], and the extent to which these individual differences reflect prior diet-induced conditioning effects is unknown. Finally, there are contradictory reports in the literature about whether LCSs elicit CPIRs in rats and humans, and it has been suggested that the differences across studies may reflect differences in the dietary history of the subjects [4].

There are several lines of support for the hypothesis that CPIR is responsive to classical conditioning. First, naïve rats generated a CPIR when consuming a solution of glucose or saccharin (an LCS), but the CPIR was eliminated when consumption of either solution was paired with gastrointestinal malaise (induced by lithium chloride injections) [25,26]. Second, when rats were maintained on a precise food-delivery schedule, they learned to secrete insulin just prior to the arrival of food [27]. Third, when rats were offered food twice a day in a hopper with a sliding door, they learned to secrete insulin in response to the sound of the door opening, irrespective of whether food was in the hopper [28].

Little is known about whether mammals can condition a CPIR to the flavor of foods or fluids. One recent study reported that hungry mice generate a CPIR immediately after initiating feeding on a familiar chow pellet [29]. While this CPIR has the hallmarks of a conditioned response to the arrival of food [28], the investigators did not explore the possibility that the CPIR reflected an unlearned response to the oral sensory properties of the chow pellet.

Here, we attempted to condition a CPIR to several highly palatable solutions, none of which elicits a CPIR in naïve mice. The test solutions contained (a) saccharin (Sacc), (b) sucrose (Suc) + acarbose, (c) maltodextrin (MD) + acarbose, or (d) flavored MD + acarbose. The addition of acarbose to the Suc and MD solutions prevented them from eliciting a CPIR in B6 mice [30]. Acarbose inhibits the enzymatic action of amylases in saliva and alpha-glucosidases on taste cells [31,32,33,34,35]. By impairing the oral digestion of Suc or MD, acarbose prevents the accumulation of free glucose in saliva. We reported previously that a solution must contain free glucose to elicit CPIR in naïve B6 mice [30].

In our conditioning paradigm, the mice consumed a training solution during five sessions. The training solutions included various concentrations of glucose, Suc or MD. Importantly, enzymes in the oral cavity and small intestine rapidly digest Suc to glucose and fructose, and MD to glucose [36,37,38]. Glucose has several postoral actions. It stimulates intestinal chemosensory cells, which activate striatal reward pathways in the brain [39,40,41] and release incretin hormones into the bloodstream [42]. Glucose is also absorbed rapidly into the bloodstream. Changes in blood glucose are sensed by beta cells in the pancreas [1] and by neurons in the hypothalamus and brainstem [43]. Stimulation of the central glucose receptors can trigger insulin secretion through a vagally mediated reflex [44]. The latter observation establishes the existence of a central mechanism by which consumption of a carbohydrate-rich solution could modulate CPIR magnitude through a classical conditioning process. We treated the flavor of the training solution as the conditioned stimulus (CS), and the postoral actions of glucose in the training solution as the unconditioned stimulus (US).

We conducted seven experiments. Experiment 1 asked whether mice would condition a CPIR to 0.61% Sacc when its consumption is paired with intragastric (IG) co-infusions of 16% glucose. Experiment 2 asked whether the consumption of a binary mixture of 0.61% Sacc + 16% glucose would condition a CPIR to 0.61% Sacc. Experiment 3 asked whether the consumption of a 32% Suc solution would condition a CPIR to 32% Suc + acarbose. Experiment 4 asked whether the consumption of a 32% MD solution would condition a CPIR to 32% MD + acarbose. In Experiments 1–4, we subjected mice to 23 h or 1 h training sessions to determine whether the effect of conditioning varied with training duration. Experiment 5 attempted to replicate the findings of the previous experiment, using a flavored 32% MD solution. Experiment 6 asked whether consumption of a lower concentration of MD (i.e., 16%) would condition a CPIR to 16% MD + acarbose. Experiment 7 asked whether the use of acarbose to block oral carbohydrate digestion altered the acceptability of the test solutions and, hence, confounded our results.

## 2. Materials and Methods

### 2.1. Animals and Housing Conditions

We used a total of 272 C57BL/6 (B6) mice that were 7–9 weeks of age, with approximately equal numbers of males and females in each treatment group. In most cases, the mice were derived from stock originally purchased from Jackson Labs (Bar Harbor, ME, USA). The one exception involved the mice that were subjected to 1 h training sessions in Experiment 1: they were purchased from Envigo (www.envigo.com). All mice weighed 20–28 g at the start of the experiments. They were housed in individual polycarbonate tub cages (27.5 × 17 × 12.5 cm) lined with Bed-O-Cobs bedding (Anderson, Maumee, OH, USA) and Enviro-dri enrichment bedding (Shepherd Specialty Papers, Framingham, MA, USA), unless indicated otherwise. The vivarium was maintained at 22 °C on a 12:12 h light/dark cycle.

The animals were provided unlimited access to chow (5001, PMI Nutrition International, Brentwood, MO, USA) and tap water, unless indicated otherwise. The chow diet contained five carbohydrates at the following concentrations (by weight): starch (31.9%), sucrose (3.7%), lactose (2.0%), fructose (0.4%), and glucose (0.2%), according to the manufacturer. The mice drank water from sipper spouts (with a 1.5 mm hole) attached to the water bottles placed through the top of the cage.

### 2.2. Chemical Stimuli

All chemical stimuli were dissolved in deionized water, prepared on the day of testing, and offered to mice at room temperature. The chemical stimuli included saccharin sodium salt hydrate (>99% pure), D-(+)-glucose (>99.5% pure), sucrose (99.5% pure), acarbose (>95% pure) (Sigma-Aldrich, St. Louis, MO, USA), and Intralipid (a stable fat emulsion; Baxter Healthcare Corporation, Deerfield, IL, USA). We used Sol Carb (Medica Nutrition, Westbury, NY, USA) as a representative maltodextrin (MD). MDs are produced by the hydrolysis of starches and are added to many processed foods. In contrast to starches, MDs are water-soluble and highly preferred by rodents [45]. The relative composition of glucose polymer lengths (reflected as degree of polymerization, DP) in SolCarb is as follows: DP 1 (glucose) = 2%, DP 2 (maltose) = 7%, DP 3 (maltotriose) = 7.8%, DP 4–7 = 43.2%, DP 8–25 = 25%, DP 26–40 = 0%, and DP > 40 = 15%. In Experiment 5, we mixed 32% MD with 0.05% (wt/wt) grape or cherry unsweetened Kool-Aid mix (General Foods, White Plains, NY, USA) to give the MD solution a more complex flavor. Kool-Aid flavors have been previously shown to be effective CSs in studies of conditioned flavor preference with mice [46].

None of the mice had any prior exposure to the training solutions in this study; thus, they were considered naïve to the flavor of the saccharin (Sacc), glucose (Gluc), sucrose (Suc), and MD solutions. While there were small quantities of sugars in the chow (~6% by weight), the sweet taste of these sugars was likely masked by the other flavorful food constituents in the chow—e.g., soybean meal, beet pulp, fish meal, ground oats, Brewer’s yeast, and alfalfa meal. The mice would have had prior exposure to milk during nursing, but mouse milk contains relatively low concentrations (<2.4%) of a single sugar (lactose) [47], and this sugar is minimally preferred by mice [48].

### 2.3. Apparatus

In Experiments 1–6, the mice were tested individually in custom-made plastic drinkometer cages (15 × 15 × 32 cm) with a stainless-steel perforated floor. Fluid was provided through one or two stainless-steel sipper spouts (orifice diameter: 1.5 mm) attached to 50 mL plastic centrifuge tubes and connected to a lick circuit (Med Associates, Fairfax, VT, USA), which tallied the number of licks. In some 23 h training sessions, the mice were housed in the drinkometers across multiple days. To access the spout, the mouse had to insert its nose through a slot (5 × 20 mm) in a stainless-steel plate at the front of the cage.

We measured fluid consumption (to the nearest 0.1 g) from each bottle by recording the change in weight over the test session with an electronic balance interfaced to a computer. Fluid spillage was estimated by recording the change in weight of bottles containing each of the different fluids in an empty cage. To correct for fluid spillage, we subtracted the estimated spill from the quantity consumed over each test session. In the 23 h training sessions, chow pellets were available in the drinkometers from a stainless-steel wire-mesh tube that entered the back wall of the cage.

In Experiment 7, we measured the relative acceptability of the test solutions in a two-bottle acceptability test by measuring the licks for each solution with a commercial gustometer (Davis Rig, www.med-associates.com).

### 2.4. Food-Restriction Schedule for the 1 h Training Sessions

To motivate consumption of the training solutions during the 1 hr training sessions, the mice were maintained on a restricted food schedule that kept them at 90% of their ad libitum body weight. To this end, we determined the weight of each mouse under ad libitum food and water conditions for several days prior to the experiment. Then, during the acclimatization stage of the experiment, we restricted the quantity of food available to each mouse by offering it variable numbers of 0.5 and 1.0 g chow pellets (Bio-Serv, Flemington, NJ, USA; bio-serv.com) until its weight stabilized at the desired weight. This process usually took five days.

It was not necessary to motivate the mice to consume the test solution during the 23 h training sessions. Thus, the mice were not subjected to a deprivation schedule during these training sessions.

### 2.5. Blood Glucose and Insulin Test (BGIT)

At the onset of each BGIT (i.e., 0 min), we obtained a blood sample to determine baseline blood glucose and plasma insulin levels (see below for details). Immediately afterwards, we put the mouse in the drinkometer with the test solution. Each mouse was required to take 200 licks. We selected this number of licks because it produces a robust and reliable CPIR in mice when the test solution contains a relatively high concentration of glucose or a glucose-containing carbohydrate [30]. Once mice completed 200 licks, the test solution was removed from the drinkometer. Most mice completed the requisite 200 licks within 1–2 min. If a mouse did not do so within 5 min, it was removed from the experiment.

The 60-min BGIT was considered to have begun once the mouse took its first lick. We collected four additional tail-blood samples 5, 15, 30, and 60 min after the BGIT began to determine the impact of the test solution on blood glucose and plasma insulin levels. All BGITs were conducted between 1:00 and 4:00 p.m. We defined CPIR as a significant rise in plasma insulin (across all mice in the treatment group) between the baseline and 5 min measurements.

Immediately prior to each BGIT, the mice were water-deprived for 23 h and food-deprived for 6 h. The water deprivation motivated mice to lick avidly for the test solution, and the food deprivation limited the quantity of food in the gastrointestinal tract during the test.

### 2.6. Tail Blood Collection, and Measurement of Blood Glucose and Plasma Insulin

During each BGIT, tail-blood samples were obtained from each mouse at five time-points (baseline, 5, 15, 30, and 60 min). To this end, the distal 1–2 mm of the tail was snipped. For blood glucose measurements, a single drop of blood was analyzed with a handheld glucometer (OneTouch Ultra; Milpitas, CA, USA). This brand of glucometer generates blood glucose levels that closely match those from a laboratory biochemical test [49]. For plasma insulin measurements, ~30 µL of blood was collected by gently stroking the tail lengthwise. These samples were collected in EDTA-coated capillary tubes (Innovative Medical Technologies; Shawnee Mission, KS, USA) and stored on ice until they were centrifuged for 7 min at 7000 rpm. The decanted plasma was stored at −80 °C until analysis with the Ultra-Sensitive Mouse Insulin ELISA (Crystal Chem; Downers Grover, IL, USA).

### 2.7. Experimental Design and Analysis

In all experiments, the experimental unit was a single mouse. We did not exclude any mice from the data analyses. The sample size for each treatment level (in a given experiment) is provided in the figure legends. All data were analyzed with Prism, v10 (graphpad.com, accessed on 17 June 2024). We evaluated the results from each treatment group (in an experiment) for normality, using the Shapiro–Wilk test. Because the results from all treatment groups passed this test (*p* > 0.05), we used parametric statistical procedures throughout the study. For the mixed-model ANOVAs, we did not assume that the results met the sphericity assumption, and we subjected the degrees of freedom to the Geisser–Greenhouse correction. For each experiment, we determined a priori (based on pilot data) the minimum sample size required to obtain a significant CPIR in each experiment, according to a one-sample *t*-test. We used an effect size of 1, power of 0.8, and alpha error probability of 0.05 [50]. Because the standard deviation of the CPIR estimates varied across the sapid solutions, the power analysis recommended different numbers of mice for each experiment.

In Experiments 1–6, the primary outcome measure was whether the mice exhibited a CPIR. The secondary outcome measures included intake during the training sessions, and changes in plasma insulin and blood glucose across the 60 min BGIT. In Experiment 7, the primary outcome measure was the lick rate.

### 2.8. Does Pairing Consumption of 0.61% Sacc with IG Co-Infusions of 16% Gluc Condition a CPIR to 0.61% Sacc? (Experiment 1)

We tested the hypothesis that pairing consumption of 0.61% (30 mM) Sacc with IG co-infusions of 16% (0.89 M) Gluc would condition a CPIR to 0.61% Sacc. The CS was the flavor of 0.61% Sacc, and the US was the postoral actions of the glucose derived from the Gluc co-infusions. We reported previously that orally consumed 0.61% Sacc (in the absence of Gluc co-infusion) does not elicit CPIR in naïve B6 mice [30].

At the onset of the experiment, all mice were fitted with an intragastric (IG) catheter. Once the mice recovered from the surgery, they were subjected to five 23 h or 1 h training sessions during which consumption of 0.61% Sacc was paired with IG co-infusions of 16% Gluc. Afterwards, each mouse was subjected to two BGITs—the first with 0.61% Sacc and the second with 16% Gluc (a positive control).

#### 2.8.1. Gastric Surgery

The mice subjected to 23 h training sessions were derived from our breeding colony. We fitted them with an intragastric catheter under isoflurane anesthesia. The catheter consisted of microrenathane tubing (0.033 in. OD × 0.014 in. ID; Braintree Scientific, Braintree, MA, USA). The tip of this tubing was heat flanged and fitted with a silastic collar that served as an anchor to keep the tube in the stomach. The tube and collar were inserted into the stomach through a small incision in the greater curvature and secured with a purse-string suture (4-0 nylon). A polypropylene mesh (7 mm^2^), which was positioned near the catheter tip, was fixed against the stomach and kept in place by a second Silastic collar. The distal end of the catheter passed through an incision in the abdominal muscle, was routed under the skin to the back of the neck, and passed through a hole in the skin. The tip of the catheter was closed with a heat seal. The abdominal incision was closed with suture (4-0 nylon), and then the skin incision was sutured closed (4-0 nylon) and treated with triple-antibiotic ointment. After the surgery, the mice were offered a diet consisting of water, regular chow, and a soft chow formulation consisting of a 16% glucose solution mixed with powdered chow for approximately 1 week. During the second week of recovery, the mice were offered regular chow and water ad libitum. After recovery, the mice were anesthetized for ~5 min with isoflurane, and the gastric catheter was extended with a 27 cm length of microrenathane tubing. The tubing passed through an infusion harness with a spring tether (CIH62; Instech Laboratories, Plymouth Meeting, PA, USA) that was fitted to the mouse. The output port of the swivel was connected to the mouse’s gastric catheter tubing, and the input port was connected to a 30 mL plastic syringe mounted in a syringe pump (A-99; Razel Scientific, Stamford, CT, USA).

The mice subjected to 1 h training sessions were purchased from Envigo and arrived pre-fitted with an intragastric (IG) catheter (https://www.inotivco.com/surgical/rat-catheterizations-options, accessed on 17 June 2024). The distal end of the catheter was routed under the skin to the mouse’s back (between the scapulae), where it was fitted to a vascular access button (VAB) attached to the skin (https://www.instechlabs.com/blog/guide-to-vascular-access-buttons, accessed on 17 June 2024). The VAB was connected to a spring tether (CIH62; Instech Laboratories, Plymouth Meeting, PA, USA), which was attached to a swivel on a counterbalanced lever (Instech Laboratories) positioned at the top of the cage.

#### 2.8.2. Training Sessions

##### 23 h Training Sessions

The mice were housed in the drinkometer with ad libitum access to chow and one bottle of water throughout the experiment. On days 1–5, they were acclimated to the harness system, regular handling, and IG infusions of water as they drank water from the sipper tube (see timeline in Appendix A). On days 6–10, the mice were given a second bottle containing 0.61% Sacc during the 23 h training sessions. The 0.61% Sacc consumption was matched 1:1 with IG co-infusions of 16% Gluc. We measured volumes of fluid ingested and infused IG at the end of the training sessions. Upon completing the training sessions, the mice were disconnected from the infusion system and provided chow and water ad libitum. On day 12, the mice were subjected to a BGIT with 0.61% Sacc; and on day 14, they were subjected to second a BGIT with 16% Gluc. We considered 16% Gluc as a positive control treatment because it elicits a CPIR in naïve mice [30].

##### 1 h Training Sessions

On days 1–5, the mice were housed in their home cages and were familiarized with having the VAB attached and detached. They were also placed in the drinkometer for 2 h/day (see timeline in Appendix A). During the first hour, they had ad libitum access to water; during the second hour, they did not have access to any fluid. Intake of the water was matched 1:1 with IG co-infusions of water. On days 6–10, we continued placing the mice in the drinkometer for 2 h/day. During the first hour, however, they had ad libitum access to 1% Intralipid; during the second hour, they did not have access to any fluids. We used 1% Intralipid because it is a stable oil emulsion that stimulates intake but has a different taste quality and postoral consequences than sweeteners [51]. Consumption of 1% Intralipid was paired with IG infusions of water. On days 11–15, the mice were subjected to the 1 h training sessions, during which they received a 0.61% saccharin solution matched 1:1 with IG infusions of 16% glucose. Following each 1 h training session, the mice were left in the cage for an additional hour, without access to any fluids. Then, they were returned to their home cage. On day 17, the mice were subjected to the first BGIT with 0.61% Sacc; and on day 19, they were subjected to second a BGIT with 16% Gluc (positive control).

On days 6–15, the mice were food-restricted to 90% of their ad libitum body weight to motivate consumption during the 1 h training sessions.

#### 2.8.3. IG Infusion Procedure during Each Training Session

The drinkometer and associated computer recorded the licks and turned the IG infusion pumps on or off, as required, every 3 s. The pump rate was 0.5 mL/min, and the oral intake-to-infusion ratio was maintained at 1:1 by adjusting a lick/pump activation parameter. Intakes were measured to the nearest 0.1 g, and IG infusions were recorded to the nearest 0.5 mL. We tested the patency of the IG catheters every day by infusing ~0.2 mL of water through them.

#### 2.8.4. Data Analysis

We used a mixed-model ANOVA to analyze plasma insulin and blood glucose dynamics across the 60 min BGITs. We calculated CPIR magnitude by subtracting plasma insulin concentration at baseline from that collected at the 5-min time-point during each of the BGITs; then, we compared CPIR magnitudes to zero with a one-sample *t*-test. We compared CPIR magnitudes elicited by saccharin versus glucose, using paired *t*-tests. Finally, we tested for a difference in blood glucose concentration between the baseline and 5-min time-points of the BGIT (using a paired *t*-test), separately, for the mice offered 0.61% Sacc or 16% Gluc.

### 2.9. Does Consumption of a Mixture of 0.61% Saccharin + 16% Glucose Condition a CPIR to 0.61% Saccharin? (Experiment 2)

We tested the hypothesis that consumption of a mixture of 0.61% Sacc + 16% Gluc (henceforth, S + G) would condition a CPIR to 0.61% Sacc. The CS was the flavor of the 0.61% Sacc solution, and the US was the postoral actions of the glucose derived from the ingested S+G solution. This experiment was similar to the previous experiment, but the 16% Gluc was ingested orally rather than infused IG during the training sessions. Owing to the presence of 16% Gluc in the S + G, it would have elicited CPIRs throughout the training sessions [30].

All mice were acclimated to drinking in the drinkometer and then randomly assigned to one of the three training sessions with the S + G solution: 23 h, 1 h, or no training (negative control). Following training, all mice were randomly assigned to one of three test solutions for the BGIT: 0.61% Sacc, 16% Gluc, or S + G. The latter two test solutions served as positive controls.

#### 2.9.1. Training Sessions

##### 23 h Training Sessions

On days 1–5, the mice were acclimated to the drinkometer and received ad libitum access to water and chow. On days 6, 8, 10, 12, and 14, the mice had ad libitum access to the S + G solution, water, and chow for 23 h (see timeline in Appendix A). On days 7, 9, 11, 13, and 15, the mice had ad libitum access to two water bottles and chow for 23 h. Fluid intakes were measured during the acclimation and training sessions. On day 17, the BGIT was conducted with one of the three test solutions.

##### 1 h Training Sessions

On days 1–5, mice were acclimated to the drinkometer for 2 h/day per day (see timeline in Appendix A). During the first hour, the mice had ad libitum access to 1% Intralipid; during the second hour, they did not have access to any fluid. On days 6 to 10, the mice received S + G solution for 1 h/day, and were left in the drinkometer for an additional hour without water. Fluid intake was measured during the acclimation and training sessions. On days 1–10, the mice were maintained on the 90% food-restriction schedule to motivate consumption during the acclimation and training sessions. On day 12, the BGIT was conducted with one of the three test solutions.

##### No Training Sessions (Negative Control)

On days 1–5, the mice were acclimated to the drinkometer for 2 h/day per day. During the first hour, the mice had ad libitum access to 1% Intralipid; during the second hour, they did not have access to any fluid. To motivate consumption of the 1% Intralipid, the mice were maintained on the 90% food-restriction schedule. Fluid intakes were measured during the acclimation sessions. On days 6–10, the mice were kept in their home cages. On day 12, the BGIT was conducted with one of the three test solutions.

#### 2.9.2. Data Analysis

We used mixed-model ANOVA to (i) determine whether the mice offered 0.61% Sacc, 16% Gluc, or S + G during the BGITs consumed similar amounts of S + G solution during the training sessions; and (ii) analyze changes in plasma insulin and blood glucose across the 60 min BGIT. We calculated CPIR magnitude by subtracting plasma insulin concentration at baseline from that collected at the 5-min time-point of the BGIT separately for tests involving Sacc, Gluc, or S + G. Then, we compared CPIR magnitudes to zero with a one-sample *t*-test. Finally, we tested for a difference in blood glucose concentration between the baseline and 5 min time-points of the BGIT (using a paired *t*-test) separately for mice offered 0.61% Sacc, 16% Gluc, or S + G.

### 2.10. Does Consumption of 32% Suc Condition a CPIR to 32% Suc + Acarbose? (Experiment 3)

We tested the hypothesis that consumption of a 32% Suc solution would condition a CPIR to 32% Suc + 5 mM acarbose. In this experiment, the CS was 32% Suc and the US was the postoral actions of the glucose derived from the ingested 32% Suc. We reported previously that 32% Suc elicits a CPIR, but 32% Suc + acarbose does not in naïve B6 mice [30].

The mice were randomly assigned to one of the three training conditions: 23 h, 1 h, or no training with 32% Suc. Afterwards, all mice were randomly assigned to one of two test solutions for the BGIT: 32% Suc + acarbose or 32% Suc (positive control).

The mice subjected to 23 h sessions were trained as in Experiment 2, except that on days 6, 8, 10, 12, and 14, they had ad libitum access to 32% Suc, water, and chow for 23 h (see timeline in Appendix A). On day 17, the BGIT was conducted with one of the two test solutions (32% Suc + acarbose or 32% Suc). The mice subjected to 1 h sessions were trained as in Experiment 2, except they received 32% Suc for 1 h/day on days 6–10. On day 12, the BGIT was conducted with 32% Suc + acarbose or 32% Suc. The mice subjected to no training sessions were treated as in Experiment 2, except that the BGIT on day 12 was conducted with 32% Suc + acarbose or 32% Suc. Data analysis was similar to Experiment 2.

### 2.11. Does Consumption of 32% MD Condition a CPIR to 32% MD + Acarbose? (Experiment 4)

We tested the hypothesis that consumption of a 32% MD solution would condition a CPIR to 32% MD + 5 mM acarbose. In this experiment, the CS was 32% MD, and the US was the postoral actions of the glucose derived from the ingested 32% MD. We reported previously that consumption of another MD (32% Polycose) elicited a CPIR, but that consumption of 32% Polycose + 5 mM acarbose did not in naïve B6 mice [30].

The mice were randomly assigned to one of the three training conditions: 23 h, 1 h, or no training sessions with 32% MD. Afterwards, the mice were randomly assigned to one of two test solutions for the BGIT: 32% MD + acarbose or 32% MD (positive control).

The mice subjected to 23 h sessions were trained as in Experiment 2 except that on days 6, 8, 10, 12 and 14 they had ad libitum access to 32% MD (see timeline in Appendix A). On day 17, each mouse was subjected to a BGIT with one of two test solutions: 32% MD + acarbose or 32% MD (positive control).

The mice subjected to 1 h sessions were trained as in Experiment 2, except they received 32% MD for 1 h/day on days 6–10. On day 12, each mouse was subjected to a BGIT with 32% MD + acarbose or 32% MD. The mice subjected to no training sessions were treated as in Experiment 2. Data analysis was similar to Experiment 3.

### 2.12. Does Consumption of Flavored 32% MD Condition a CPIR to Flavored 32% MD + Acarbose? (Experiment 5)

This experiment had two goals. The first was to replicate the main 23 h findings of the previous experiment. The second was to determine whether supplementing the flavor of the MD solution would enhance the magnitude of the conditioned CPIR, relative to what we observed in Experiment 4. That latter possibility was based on a prior report that salient odors enhance the conditioned hypoglycemic response to oral stimuli [52]. To this end, we added unsweetened 0.05% grape or cherry Kool-Aid to the 32% MD solution (henceforth, flavored 32% MD). Approximately equal numbers of mice in each treatment group received the grape or cherry flavor.

In this experiment, the CS was the flavored 32% MD solution, and the US was the postoral actions of the glucose derived from the ingested flavored 32% MD solution. Because we demonstrated in the previous experiment that mice lacking the five 23 h training sessions with 32% MD did not exhibit a CPIR to 32% MD + acarbose, we did not subject mice to the no-training regime in the present experiment. The mice were trained as the 23 h mice in Experiment 4, except that on days 6, 8, 10, 12, and 14, they had ad libitum access to flavored 32% MD solution (see timeline in Appendix A). On day 17, each mouse was subjected to a BGIT with one of two test solutions: flavored 32% MD + acarbose or flavored 32% MD (positive control). The mice were randomly assigned to the test solutions. Data analyses were as in the prior experiment.

### 2.13. Does Consumption of 16% MD Condition a CPIR to 16% MD + Acarbose? (Experiment 6)

This experiment sought to replicate the 23 h findings from Experiment 4, except using a 16% MD solution. The CS was the flavored 16% MD solution, and the US was the postoral actions of the glucose derived from the ingested 16% MD solution. We hypothesized that the 16% MD solution would condition a CPIR to 16% MD + 5 mM acarbose. The mice were trained as in Experiment 4, except that on days 6, 8, 10, 12, and 14, they had ad libitum access to 16% MD (see timeline in Appendix A). On day 17, each mouse was subjected to a BGIT with one of two test solutions: 16% MD + acarbose or 16% MD (positive control). The mice were randomly assigned to these test solutions and to the exposure treatments. The mice subjected to no training sessions were treated as in Experiment 4, but the BGIT on day 12 was conducted with 16% MD + acarbose or 16% MD. Data analysis was similar to Experiment 4.

### 2.14. Does Acarbose Alter the Acceptability of Water, 32% Suc or 32% MD? (Experiment 7)

In Experiments 3–6, we used 5 mM acarbose to block oral carbohydrate digestion. In so doing, we assumed that the acarbose did not alter the acceptability of the test solutions. If it did alter acceptability, then the acarbose would have confounded our results. To test this assumption, we used a two-bottle acceptability test to compare short-term licking responses of naïve mice to (i) water vs. acarbose, (ii) 32% Suc vs. 32% Suc + acarbose, and (iii) 32% maltodextrin (MD) vs. 32% MD + acarbose.

#### 2.14.1. Two-Bottle Acceptability Test

Prior to the lick tests, all mice were subjected to three training sessions in the gustometer. This training habituated the mice to the gustometer and trained them to obtain water from the sipper tube. Each training session began when a mouse took its first lick, and lasted 30 min; during this time, the mouse could drink water freely from a single sipper tube. To motivate licking, the mouse was water-deprived for 22.5 h prior to the first training session. Afterwards, the mouse was returned to its home cage and given 1 h of ad libitum access to water and food; then, it was water-deprived for another 22.5 h. Once training was complete, the mouse received at least one recovery day.

After completing training, we subjected each mouse to a 30 min test, during which each mouse was presented two solutions during consecutive 5 s trials. During each 5 s trial, however, the mouse could only access one of the solutions. We treated the two test solutions as a block and randomized (without replacement) their presentation sequence within each block. The mouse could initiate as many 5 s trials (and hence, blocks) as possible during the test session. Prior to testing water vs. acarbose, the mice were subjected to 23 h of water deprivation. Prior to testing the other two pairs of solutions, the mice were subjected to 23 h of food and water restriction (i.e., 1 g chow and 2 mL of water).

#### 2.14.2. Data Analysis

For each two-bottle acceptability test, we calculated the mean number of licks per trial across the 30 min test session, separately, for each test solution. Then, we used paired *t*-tests to compare mean lick rates for (i) water vs. acarbose, (ii) 32% Suc vs. 32% Suc + acarbose, and (iii) 32% MD vs. 32% MD + acarbose.

## 3. Results

### 3.1. Does Matching Consumption of 0.61% Sacc with IG Infusions of 16% Gluc Condition a CPIR to 0.61% Sacc? (Experiment 1)

The ANOVA confirmed that the mice obtained equivalent amounts of 0.61% Sacc (orally) and 16% glucose (IG) during the 23 h (Figure 1A) and 1 h (Figure 1B) training sessions (Appendix A). Furthermore, a visual inspection of the graphs reveals that the mice obtained substantially more of each solution during the 23 h (11–15 g) than the 1 h (~2 g) training sessions.

Upon completing the training sessions, the mice were subjected to two BGITs—one with 0.61% Sacc and the other with 16% Gluc. The ANOVA revealed an interaction of test solution × time across the BGIT for both plasma insulin and blood glucose levels in mice subjected to 23 h (Figure 1C,G) and 1 h (Figure 1D,H) training sessions (Appendix A). These significant interactions reflect the fact that plasma insulin and blood glucose levels increased during the BGIT with 16% Gluc (peaking at 15 min), but not during the BGIT with 0.61% Sacc.

0.61% Sacc failed to elicit a CPIR (i.e., an increase in plasma insulin, relative to baseline, during the initial 5 min of the test) in mice subjected to 23 h or 1 h training sessions. In contrast, 16% Gluc elicited a robust CPIR, irrespective of training session duration (Figure 1E,F). Furthermore, the magnitude of the insulin response elicited by 16% Gluc at the 5-min time-point was larger than that elicited by 0.61% Sacc (Figure 1E,F).

Blood glucose levels increased significantly above baseline by the 5-min time-point of the BGIT with 16% Gluc but not 0.61% Sacc. This result was observed in mice subjected to 23 h (Figure 1G) or 1 h (Figure 1H) training sessions.

In sum, these results contradict the hypothesis that pairing consumption of 0.61% Sacc with IG co-infusions of 16% Gluc would condition a CPIR to the flavor of 0.61% Sacc.

### 3.2. Does Consumption of S + G Condition a CPIR to 0.61% Sacc? (Experiment 2)

We show the amounts of S + G consumed during the 23 h (Figure 2A) and 1 h (Figure 2B) training sessions. Within each panel, we distinguish consumption by mice that were later subjected to a BGIT with 0.61% Sacc, 16% Gluc, or S + G. An ANOVA confirmed that the mice in the different treatment groups consumed similar amounts of S + G across the five 23 h and 1 h training sessions (Appendix A). Even though consumption increased across the 23 h and 1 h training sessions, the interaction of training session × test solution was nonsignificant. Finally, a visual inspection of the results revealed that the mice consumed substantially more S + G during the 23 h (10–12 g) than the 1 h (~2 g) training sessions.

Once the training sessions ended, the mice were subjected to a BGIT with 0.61% Sacc, 16% Gluc, or S + G. The ANOVA revealed an interaction of test solution × time across the BGIT for both plasma insulin and blood glucose levels, irrespective of whether the mice were subjected to 23 h (Figure 2C,I), 1 h (Figure 2D,J), or no training (Figure 2E,K) sessions with S + G (Appendix A). The significant interactions reflect the fact that plasma insulin and blood glucose levels increased markedly following consumption of 16% Gluc and S + G (peaking at 5–15 min) but did not change following consumption of 0.61% Sacc.

Consumption of 0.61% Sacc failed to elicit a CPIR following all three training conditions (Figure 2F–H). Unexpectedly, the 0.61% Sacc actually caused plasma insulin levels to drop slightly (but significantly) below baseline in mice subjected to either 1 h or no training sessions with S + G (in both cases, *p* < 0.04). In contrast, licking for the positive control solutions (i.e., 16% Gluc or S + G) elicited robust CPIRs, irrespective of training session duration. Indeed, the magnitude of the insulin response elicited by 16% Gluc or S + G at the 5-min time-point was significantly larger than that elicited by 0.61% Sacc (Dunnett’s multiple comparison test, *p* < 0.05).

Blood glucose levels increased significantly above baseline by the 5-min time-point of the BGIT in mice offered 16% Gluc or S + G, but not in mice offered 0.61% Sacc. This result was observed in mice subjected to 23 h (Figure 2I), 1 h (Figure 2J), or no (Figure 2K) training sessions with the S + G solution.

Taken together, these results contradict the hypothesis that consumption of S + G would condition a CPIR to the flavor of 0.61% Sacc.

### 3.3. Does Consumption of 32% Suc Condition a CPIR to 32% Suc + Acarbose? (Experiment 3)

We show the amounts of 32% Suc solution consumed during the 23 h (Figure 3A) and 1 h (Figure 3B) training sessions. Within each panel, we distinguish consumption by mice that were later subjected to a BGIT with either 32% Suc + acarbose or 32% Suc. An ANOVA confirmed that mice in the different BGIT treatment groups consumed similar amounts of MD solution during the training sessions (Appendix A). Furthermore, even though consumption decreased across the 23 h (and increased across the 1 h) training sessions, the interaction of training session × test solution was not significant in both analyses. A visual inspection of the graphs reveals, however, that the mice ingested considerably more 32% Suc during the 23 h (8–9 g) than the 1 h (1.5–2.5 g) training sessions.

Once the training sessions ended, the mice were subjected to a BGIT with 32% Suc + acarbose or 32% Suc alone. The ANOVA revealed an interaction of test solution × time across the BGIT for both plasma insulin and blood glucose levels in mice subjected to 23 h (Figure 3C,I), 1 h (Figure 3D,J), or no (Figure 3E,K) training sessions with the 32% Suc solution (Appendix A). For plasma insulin, the interactions reflect the fact that plasma insulin levels rose above baseline during the BGIT with 32% Suc (peaking at 15 min), but did not rise during the test with Suc + acarbose, irrespective of training session duration. For blood glucose, the interactions reveal that blood glucose levels increased above baseline in both groups after 5 min, but they did so to a much greater extent in mice offered 32% Suc. This latter finding confirms that the acarbose reduced, but did not eliminate, digestion of sucrose.

The 32% Suc + acarbose caused insulin levels to drop slightly (but significantly) below baseline in mice subjected to 23 h (Figure 3F) or no (Figure 3H) training sessions with 32% Suc. There was no analogous drop in insulin levels in mice subjected to 1 h training sessions (Figure 3G). In contrast, 32% Suc triggered a robust CPIR in mice subjected to 23 h, 1 h, or no training sessions with 32% Suc. Under each training condition, the magnitude of the insulin response to 32% Suc at the 5-min time-point was significantly higher than that to 32% Suc + acarbose (in all cases, *p* < 0.05; unpaired *t*-test).

Blood glucose levels increased significantly above baseline at the 5-min time-point of the BGIT with 32% Suc + acarbose and 32% Suc. This result was apparent in mice subjected to 23 h (Figure 3I), 1 h (Figure 3J), or no (Figure 3K) training sessions with 32% Suc.

In sum, these results contradict the hypothesis that consumption of 32% Suc conditions a CPIR to the flavor of 32% Suc + acarbose.

### 3.4. Does Consumption of 32% MD Condition a CPIR to 32% MD + Acarbose? (Experiment 4)

We show the amounts of 32% MD consumed during the 23 h (Figure 4A) and 1 h (Figure 4B) training sessions. Within each panel, we distinguish consumption by mice that were later subjected to a BGIT with either 32% MD + acarbose or 32% MD (the positive control solution). A mixed-model ANOVA confirmed that mice in the different BGIT treatments consumed similar amounts 32% MD during the training sessions (Appendix A). While consumption was steady across the 23 h (and increased across the 1 h) training sessions, the interaction of training session × test solution was not significant in both analyses. A visual inspection of the data illustrates that the mice ingested considerably more 32% MD during the 23 h (8–9 g) than during the 1 h (1–3 g) training sessions.

Once the training sessions ended, the mice were subjected to a BGIT with 32% MD + acarbose or 32% MD. A mixed-model ANOVA revealed an interaction of test solution × time across the BGIT for plasma insulin levels in mice with 23 h (Figure 4C), 1 h (Figure 4D), or no (Figure 4E) training sessions with 32% MD (Appendix A). For mice with 23 h of training with 32% MD, plasma insulin levels rose above baseline during the BGITs with both 32% MD and 32% MD + acarbose, but they rose to a greater extent during the BGIT with 32% MD. In contrast, for mice subjected to 1 h or no training sessions with 32% MD, plasma insulin levels rose above baseline during the BGIT with 32% MD, but not during the BGIT with 32% MD + acarbose.

There was an interaction of test solution × time across the BGIT for blood glucose levels in mice that received 23 h (Figure 4I), 1 h (Figure 4J), or no (Figure 4K) training sessions with 32% MD (Appendix A). The significant interaction reflects the fact that blood glucose levels increased above baseline during the BGITs with both 32% MD and 32% MD + acarbose, but they rose to a greater extent during the BGIT with 32% MD in all training regimens. This latter observation confirms that the acarbose reduced, but did not eliminate, digestion of 32% MD.

The treatment with 32% MD + acarbose elicited a CPIR in mice subjected to 23 h training sessions with 32% MD (Figure 4F), but not in mice subjected to 1 h or no training sessions with 32% MD (Figure 4G,H). In contrast, 32% MD reliably elicited a CPIR in the mice, irrespective of training treatment (Figure 4F–H). In mice subjected to 23 h training sessions, the CPIR magnitude elicited by 32% MD + acarbose was indistinguishable from that elicited by 32% MD alone (t = 1.4, df = 15, *p* = 0.17). In mice subjected to the 1 h or no training sessions with 32% MD, the CPIR magnitudes elicited by 32% MD were larger than those elicited by 32% MD + acarbose (in both cases, t > 4.66, *p* < 0.0001).

Blood glucose levels rose significantly above baseline at the 5-min time-point of the BGIT in mice tested with 32% MD + acarbose or 32% MD. This result was apparent in mice subjected to 23 h (Figure 4I), 1 h (Figure 4J), or no (Figure 4K) training sessions with 32% MD.

These results support the hypothesis that the consumption of 32% MD conditions a CPIR to the flavor of 32% MD + acarbose; however, the effect was limited to mice subjected to 23 h training sessions.

### 3.5. Does Consumption of Flavored 32% MD Enhance the Conditioned CPIR to Flavored 32% MD + Acarbose? (Experiment 5)

We show the amounts of flavored 32% MD consumed during the 23 h (Figure 5A) training sessions. We distinguish consumption by mice that were later subjected to a BGIT with either flavored 32% MD + acarbose or flavored 32% MD. A mixed-model ANOVA confirmed that mice in both test solution treatments consumed similar amounts of flavored 32% MD during the training sessions (Appendix A). While consumption varied across the 23 h training sessions, the interaction of training session × test solution was nonsignificant in both analyses. The mice consumed 9–12 g of flavored 32% MD solution during the 23 h training sessions.

Once the training sessions ended, the mice were subjected to a BGIT with flavored 32% MD + acarbose or flavored 32% MD. A mixed-model ANOVA revealed an interaction of test solution × time across the BGIT for plasma insulin levels (Figure 5B and Appendix A). While plasma insulin levels rose above baseline during the BGITs with the flavored 32% MD and the flavored 32% MD + acarbose, they did so to a greater extent with the flavored 32% MD.

We discovered that the flavored 32% MD + acarbose and the flavored 32% MD both elicited a robust CPIR (Figure 5C). The magnitude of these CPIRs was indistinguishable (t = 1.6, df = 17, *p* = 0.14).

There was an interaction of test solution × time across the BGIT for blood glucose levels (Figure 5D and Appendix A). This interaction reflects the fact that blood glucose levels increased above baseline during the BGITs with flavored 32% MD and flavored 32% MD + acarbose, but they did so to a greater extent with the flavored 32% MD. This latter observation reveals that acarbose reduced, but did not eliminate, digestion of the 32% MD. Finally, blood glucose levels increased significantly above baseline at the 5 min time-point of the BGIT with flavored 32% MD + acarbose and flavored 32% MD.

Taken together, these results provide support for the hypothesis that 23 h of training with flavored 32% MD conditions a CPIR to flavored 32% MD + acarbose. However, contrary to expectation, the conditioned CPIR to flavored 32% MD + acarbose was not larger than that elicited by the MD + acarbose solution (in the prior experiment). If anything, the magnitude of the conditioned CPIR to flavored 32% MD + acarbose (mean = 0.18 ng/mL) tended to be smaller than that to 32% MD + acarbose (mean = 0.31 ng/mL), based on an unpaired *t*-test (t = 1.77, df = 16, *p* = 0.10).

### 3.6. Does Consumption of 16% MD Condition CPIR to 16% MD + Acarbose? (Experiment 6)

We show the amounts of 16% MD consumed during the 23 h training sessions in Figure 6A. We distinguish consumption by the mice that were later subjected to a BGIT with either 16% MD + acarbose or 16% MD. A mixed-model ANOVA confirmed that mice in both treatment groups consumed similar amounts of 16% MD solution during the training sessions (Appendix A).

Once the training sessions ended, the mice were subjected to a BGIT with 32% MD + acarbose or 16% MD. A mixed-model ANOVA revealed an interaction of test solution × time across the BGIT for plasma insulin levels in mice with 23 h (Figure 6B) or no (Figure 6C) training sessions with 16% MD (Appendix A). In both treatment groups, plasma insulin levels rose above baseline during the BGIT with 16% MD, but not during the BGIT with 16% MD + acarbose.

The 16% MD + acarbose solution failed to elicit a CPIR in mice subjected to either 23 h or no training sessions with 16% MD (Figure 6D,E). In contrast, 16% MD reliably elicited a CPIR in mice subjected to 23 h or no training sessions.

There was an interaction of test solution × time across the BGIT for blood glucose levels in mice subjected to 23 h (Figure 6F) or no (Figure 6G) training sessions with 16% MD (Appendix A). The interaction reflects the fact that blood glucose levels rose above baseline during the BGITs with both 16% MD + acarbose and 16% MD in both groups, but they rose to a greater extent during the BGIT with 16% MD. This latter observation confirms that the acarbose reduced, but did not eliminate, digestion of 16% MD.

Blood glucose levels rose significantly above baseline at the 5-min time-point of the BGIT with 16% MD + acarbose and 16% MD. This result was observed in mice subjected to 23 h (Figure 6F) or no (Figure 6G) training sessions with 16% MD.

These results contradict the hypothesis that consumption of 16% MD would condition a CPIR to 16% MD + acarbose.

### 3.7. Why Did the Mice Condition a CPIR Exclusively to the Flavored and Unflavored 32% MD Solutions in Experiments 1–6?

We tested the hypothesis that the mice conditioned a CPIR to the flavored and unflavored 32% MD solutions because they alone caused sufficient elevations in blood glucose during the 23 h training sessions. If so, then we predicted that the mice would have obtained greater quantities of glucose from the flavored and unflavored 32% MD solutions than they did from the other test solutions.

In Figure 7A, we show the mean intake of the different test solutions during the 23 h training sessions in Experiments 1–6 (F-ratio = 95.7, df = 5, 95, *p* < 0.0001). The mice obtained about twice as much fluid when oral intake of 0.61% Sacc was matched with IG co-infusion of 16% Gluc in Experiment 1 than they did when they consumed the other test solutions in Experiments 2–6. Furthermore, the mice consumed more S + G, 16% MD, and flavored 32% MD than they did 32% Suc and 32% MD. The interpretation of these results is complicated, however, by the different composition of the test solutions. To address this complication, we estimated the amount of glucose that the mice would have obtained from each test solution across the 23 h training sessions. For these calculations, we made three simplifying assumptions: (i) the ingested MD was digested completely into free glucose molecules by amylases and alpha-glucosidases in the oral cavity and small intestine, (ii) the ingested sucrose was digested completely into free glucose and fructose molecules by alpha-glucosidases in the oral cavity and small intestine; and (iii) none of the ingested fructose (released during the digestion of sucrose) was converted to glucose in the small intestine. While the last assumption was certainly violated to some extent [53], it is unlikely that a large percentage of the dietary fructose was converted to glucose. This is because the conversion process saturates at fructose loads > 1 g/kg in B6 mice [53]. Given that the mice obtained an average of 1.35 g of fructose per 23 h training session in Experiment 3, it follows that a 25 g mouse would have experienced a fructose load of 54.2 g/kg across the 23 h training session (or 2.4 g/kg per h during the training session).

In Figure 7B, we show the estimated amounts of glucose that mice would have obtained from the different test solutions during the 23 h test sessions. These estimates indicate that the mice obtained significantly more glucose from the flavored and unflavored 32% MD solutions than they did (i) from the S + G, 32% Suc or 16% MD solutions (F-ratio = 104.5, df = 5, 95, *p* < 0.0001), or (ii) when the intake of 0.61% Sacc was matched with IG co-infusions of 16% Gluc. These results support the prediction that the mice obtained more glucose from the flavored and unflavored 32% MD solutions than they did from the other test solutions across the 23 h training sessions.

### 3.8. Does Acarbose Alter the Acceptability of Water or the Carbohydrate Solutions? (Experiment 7)

The two-bottle acceptability tests examined the impact of adding 5 mM acarbose to water, 32% Suc, or 32% MD. We did not observe any differences in lick rate for (i) water vs. water + acarbose (Figure 8A; paired t-value = 1.13, df = 7, *p* = 0.29), (ii) 32% Suc vs. 32% Suc + acarbose (Figure 8B; paired t-value = 1.0, df = 7, *p* = 0.35), or (iii) 32% MD vs. 32% MD + acarbose (Figure 8C; paired t-value = 0.7, df = 7, *p* = 0.51). These results demonstrate that the added acarbose did not alter the acceptability of water, 32% Suc, or 32% MD.

## 4. Discussion

### 4.1. The Mice Did Not Condition a CPIR to 0.61% Sacc or 32% Suc + Acarbose

In Experiments 1 and 2, we attempted to condition a CPIR to the flavor of 0.61% Sacc in two ways. We either paired the intake of 0.61% Sacc with IG co-infusions of 16% Gluc or exposed mice to the S + G solution. Neither approach conditioned a CPIR to the flavor of 0.61% Sacc, however. In Experiment 3, we tried to condition a CPIR to the flavor of the 32% Suc + acarbose solution by exposing mice to 32% Suc. This approach also failed to condition a CPIR to 32% Suc + acarbose. Three features of our study design support the inference that these negative findings are real. First, the positive control stimuli in Experiments 1 (16% Gluc), 2 (16% Gluc and S + G), and 3 (32% Suc) all elicited robust CPIRs following the 23 h and 1 h training sessions. This demonstrates that the experimental manipulations did not impair the ability of the mice to generate a CPIR. Second, the negative control solutions in Experiments 2 (0.61% Sacc), 3 (32% Suc + acarbose), 4 (32% MD + acarbose), and 6 (16% MD + acarbose) all failed to elicit a CPIR in the mice with no training session. This shows that the experimental manipulations did not cause the mice to respond to the test solutions in an aberrant manner. Third, we confirmed that the acarbose treatment was effective. It not only prevented the 32% Suc and MD solutions from eliciting a CPIR in the unexposed control mice, but it also inhibited digestion of the 32% Suc and MD solutions, as indicated by attenuated glycemic responses during the BGITs.

It is remarkable that the 23 h or 1 h training sessions in the present study failed to condition a CPIR to the flavor of 0.61% Sacc, 16% MD + acarbose, or 32% Suc + acarbose. This is because previous studies with B6 mice reported that 1 h training sessions with 4–16% glucose or 16% sucrose solutions were sufficient to condition flavor–nutrient learning in mice [54,55,56,57]. In some cases, the mice conditioned an increase in lick rate within a single 1 h training session. Our results thus indicate that it takes greater amounts of postoral glucose stimulation to condition a CPIR than it does to condition a preference for a flavored solution.

### 4.2. The Mice Conditioned a CPIR Exclusively to the 32% MD + Acarbose Solutions

We found that 23 h training sessions with the 32% MD solutions effectively conditioned a robust CPIR to the 32% MD + acarbose solutions, respectively. The fact that this result was replicated in two experiments provides strong evidence that it is real. Given that the magnitude of the conditioned CPIR in both experiments was statistically indistinguishable, we infer that the addition of the fruity flavor in Experiment 5 did not enhance the conditioning process. Finally, it is notable that 1 h training sessions with the unflavored 32% MD solution did not condition a CPIR to the unflavored 32% MD + acarbose solution. This indicates that the 1 h training sessions did not provide sufficient exposure to the 32% MD solution.

Why did the mice condition a CPIR exclusively to the 32% MD + acarbose solutions? Figure 7 reveals that, following digestion, the mice would have obtained more glucose from the flavored and unflavored 32% MD solutions than from the other test solutions. It follows that the glucose derived from the 32% MD solutions provided enough postoral glucose stimulation to induce conditioning, whereas the glucose derived from the other test solutions did not. This inference leads to the prediction that mice should be able to condition CPIR to other cephalic stimuli (e.g., 0.61% Sacc), as long as their intake is associated with postoral stimulation with 32% glucose or MD solutions. Likewise, mice should be able to condition CPIR to cephalic stimuli associated with foods that contain high concentrations of starch, although the minimum starch concentration has not yet been determined.

Two other studies provide evidence that dietary glucose can modulate CPIR magnitudes. First, ad libitum consumption of 11% glucose solutions over 4 weeks increased CPIR magnitude in B6 mice, whereas ad libitum consumption of 11% high-fructose solutions (~ 55% fructose and 40–45% glucose) over 4 weeks had no impact on CPIR magnitude [58]. Second, to simulate high intakes of glucose, Teff et al. [59] provided human subjects with continuous intravenous infusions of glucose over two days. They found that the infusions increased CPIR magnitude. We should note that these previous studies differed from the present study in an important way. They reported that elevating glucose levels in the diet or the blood enhanced CPIR magnitude. Here, we found that the training sessions with 32% MD solutions conditioned a CPIR to an oral stimulus that previously did not elicit a CPIR (i.e., 32% MD + acarbose).

### 4.3. Contribution of the Three Taste Signaling Pathways for Carbohydrates to the Conditioning Process

There are three postulated taste signaling pathways for carbohydrates. The canonical sweet-taste pathway is activated when sugars or LCSs bind to the T1R2 + R3 taste receptor [60]. The activation of this pathway mediates concentration-dependent increases in licking for sugars and sweeteners [61,62,63], but it does not mediate the ability of glucose to elicit CPIR in naïve mice [19]. The second signaling pathway is activated exclusively by glucose. It is thought to resemble the glucose-specific signaling pathway in beta cells of the pancreas [4,64,65,66,67]. The stimulation of this pathway in taste cells is necessary for oral stimulation with glucose or glucose-containing carbohydrates to trigger CPIR in naïve B6 mice [30]. The third signaling pathway is activated by water-soluble polysaccharides (i.e., MDs). Although the nature of the MD taste receptor is unknown, we know that (a) rats perceive MDs as having a distinct taste quality from sucrose [68]; (b) genetic deletion of one or both subunits of the T1R2 + R3 taste receptor in mice has little or no impact on licking responses to MD solutions [63,69], but genetic deletion of TRPM5 or alpha-gustducin attenuates MD intake and preference in mice [70]; (c) oral detection of MD + acarbose solutions by humans is not affected by blocking the T1R2 + R3 signaling pathway (e.g., with lactisole) [71]; and (d) MD + acarbose solutions elicit a “starchy” taste quality in humans which is distinct from the sweet-taste quality elicited by sugars and LCSs [71].

The results of this study offer several insights into how each of the aforementioned taste signaling pathways contributes to CPIR. First, training with a selective ligand of the T1R2 + R3 signaling pathway (i.e., Sacc) had no impact on the ability of glucose to elicit CPIR. Second, we confirmed a prior report that the stimulation of the glucose-specific taste signaling pathway reliably elicits CPIR in naïve (i.e., untrained) mice [19]. Third, the stimulation of the MD pathway with unflavored 32% MD + acarbose did not elicit a CPIR in naïve mice, but it did elicit a CPIR following the 23 h training sessions with 32% (but not 16%) MD solutions. This shows that the stimulation of the MD pathway can elicit CPIR, but only after it has been conditioned to do so with 32%, but not 16%, concentrations of MD. Fourth, the mice did not condition a CPIR to any solution that selectively stimulated the T1R2 + R3 signaling pathway (i.e., Sacc or 32% Suc + acarbose). It is possible, however, that ligands of this latter signaling pathway could be conditioned to generate a CPIR if their flavor was associated with intake of a higher concentration of glucose than was used herein (i.e., 32% Gluc).

There is a discrepancy between the CPIR magnitudes elicited by the MD used herein (SolCarb) and the one used in a prior study (Polycose) [30]. The CPIR magnitudes elicited by 16 and 32% SolCarb were substantially larger than the CPIR elicited by 32% Polycose. This discrepancy could reflect differences in composition of the MDs. For example, the % of glucose polymers with degrees of polymerization ≥ 26 is 15% for SolCarb, and 23% for Polycose. Furthermore, the sodium content of SolCarb is 70 mg/100 g, and that of Polycose is 150 mg/100 g [72]. More work is needed to determine whether MDs with different degrees of polymerization or mineral compositions elicit different CPIR magnitudes.

### 4.4. Rapid Elevations in Blood Glucose Levels Are Not Sufficient to Trigger a CPIR

To establish that an oral stimulus elicits a CPIR, it is conventional to demonstrate that the rise in plasma insulin precedes the rise in blood glucose. This is because elevations in blood sugar alone are sufficient to elicit insulin secretion [1]. To this end, we demonstrated previously that when B6 mice take a body mass-specific number of licks from a 36% glucose solution (resulting in a dose of 2 mg/g), their insulin levels rise above baseline in 3 min, while their blood glucose levels do so in 4 min [30].

We have also obtained evidence, however, that a rapid rise in blood glucose is not sufficient to trigger insulin secretion during the initial 5 min of a BGIT. For example, when we administered glucose IG (via oral gavage), blood glucose levels rose above baseline within 5 min, but insulin levels did not change over the same time period in B6 mice [21,73]. Likewise, atropine-treated mice generated a robust rise in blood glucose levels but no corresponding rise in plasma insulin levels during a 5 min BGIT with an 18% glucose solution [21]. In the present study, consumption of 32% Suc + acarbose increased blood glucose within 5 min, but it did not increase plasma insulin in mice subjected to 23 h, 1 h, or no training sessions with 32% Suc (Figure 3). Furthermore, consumption of 16 or 32% MD + acarbose increased blood glucose within 5 min, but it did not increase insulin in mice subjected to 1 h or no training sessions with the MD solutions (Figure 4 and Figure 6).

The results of the present and prior studies thus establish that consumption of glucose can elicit a rapid rise in plasma insulin prior to any increase in blood glucose in B6 mice. However, the reverse is not true. That is, a rapid rise in blood glucose is not sufficient to trigger a rapid rise in plasma insulin during the initial 5 min of a BGIT. It follows that the CPIR measurements in the present study were not confounded by the rapid elevations in blood glucose that we observed.

### 4.5. Acarbose Did Not Alter the Acceptability of the Test Solutions

The presence of acarbose did not alter the acceptability of water, 32% Suc, or 32% MD in the brief-access acceptability tests. This observation has several implications. First, the lack of discrimination between the acarbose and water solutions indicates that acarbose did not impart an off-taste, which rendered the test solutions less acceptable. Second, even though acarbose elicits a sweet taste in humans [74], it did not increase the acceptability of the water, 32% Suc, or MD solutions to the mice. This implies that the acarbose did not alter the sweet intensity of the test solutions. Third, it was reported that disaccharide solutions elicit larger peripheral taste nerve responses than the same disaccharide solutions treated with α-glucosidase inhibitors [34]. The authors inferred that, in the absence of the inhibitor, the α-glucosidases would have enzymatically liberated some fraction of the monosaccharides and thereby increased the molar concentration (and hence sweet intensity) of the sugar solutions. Our behavioral findings do not provide support the latter inference. Taken together, these findings indicate that the use of acarbose did not alter the acceptability of the 32% Suc or the flavored and unflavored 32% MD solutions.

### 4.6. Consumption of Several Test Solutions Caused a Small Decline in Plasma Insulin at the 5-Min Time-Point

In several instances, the CS solution caused a small (but significant) drop in plasma insulin levels, relative to baseline, during the initial 5 min of the BGIT. This happened during BGITs with the Sacc solution in mice that received either 1 h or no training with the S + G solution. It also happened during BGITs with 32% Suc + acarbose in mice that received either 23 h or no training with 32% Suc. The rapid declines in plasma insulin levels were not associated with any change in blood glucose levels. For example, blood glucose levels increased during the initial 5 min of the BGIT with 32% Suc + acarbose but did not change during the initial 5 min of the BGIT with 0.61% Sacc. Additional work is needed to explain these observations. Given the small size and inconsistency of these declines in plasma insulin levels, however, it is not clear whether they are functionally significant.

### 4.7. Mice Display Multiple Mechanisms for Adapting to Diets High in Glucose

We reported previously that dietary exposure to some, but not all, carbohydrates changes ingestive and metabolic responses of B6 mice. For example, we discovered that chronic consumption of high-glucose diets increased glucose tolerance, but consumption of high-fructose diets did not [58]. The high-glucose diet improved glucose tolerance by increasing both CPIR magnitude and insulin sensitivity. The present study reveals another way that mice adapt to substances that contain high concentrations of glucose. They condition a CPIR to their flavor, and thus facilitate postoral processing of the carbohydrates. 

### 4.8. Strengths and Limitations of Study Design

The strengths of the study design include the inclusion of multiple positive and negative control treatments and the use of 23 h and 1 h training sessions. All of these experimental design features facilitated interpretation of the findings. Furthermore, the fact that the mice conditioned a CPIR to the unflavored or flavored 32% MD + acarbose solutions establishes that our conditioning protocol was effective. These positive findings also add credibility to the negative findings we obtained in Experiments 1, 2, 3, and 6.

Our study design had several limitations. We used relatively high concentrations of sugars and MDs (i.e., 16–32%) in our training and test solutions to reduce the chances of committing a type-2 error. Although most commercial beverages have lower sugar concentrations (i.e., 10–12%; [75,76]), many raw and processed foods have carbohydrate (i.e., sugar + starch) concentrations that exceed 50% of the total caloric content [77] (https://ciqual.anses.fr/, accessed on 20 June 2024). These high-carbohydrate foods would be expected to produce postoral glucose stimulation in much the same way as the training solutions used herein. Second, we trained mice with the S + G solution in Experiment 2. We did so to determine whether the mice could condition a CPIR to the flavor of Sacc when it was associated with both oral stimulation of the glucose-specific taste signaling pathway and the elicitation of CPIRs. The limitation of this approach is that we did not know the extent to which the flavor of the S + G solution generalized to the flavor of the Sacc solution during the BGIT.

## 5. Conclusions

We tested the hypothesis that mice can condition a CPIR to the flavor of sapid solutions that cause postoral glucose stimulation. To this end, we tested six different sapid solutions, none of which elicited a CPIR in untrained mice. Whereas the mice conditioned a CPIR to the 32% MD + acarbose solutions, they did not do so to the other sapid solutions. We propose that the mice conditioned a CPIR exclusively to the 32% MD + acarbose solutions because the 32% MD training solutions alone provided sufficient postoral glucose stimulation. This inference is best supported by the observation that the mice conditioned a CPIR to 32% MD + acarbose but not to 16% MD + acarbose. The only difference between the training solutions was a 2-fold difference in glucose concentration, following digestion of MD in the mouth and small intestine.

Future studies should determine whether mice can condition a CPIR to the flavor of 0.61% Sacc when its intake is associated with higher levels of postoral glucose stimulation—e.g., from a 32% glucose or MD solution. Such a finding would provide support for current concerns that regular consumption of calorically dense foods/beverages containing LCSs, sugars, and/or maltodextrins alters insulin responses to sweet or sugary foods. It may also provide insight into why LCSs have been reported to elicit a CPIR in some but not all studies involving rat and human subjects. It is possible, for instance, that subjects in some of the studies had previously conditioned a CPIR to an LCS.

## Figures and Tables

**Figure 1 nutrients-16-02250-f001:**
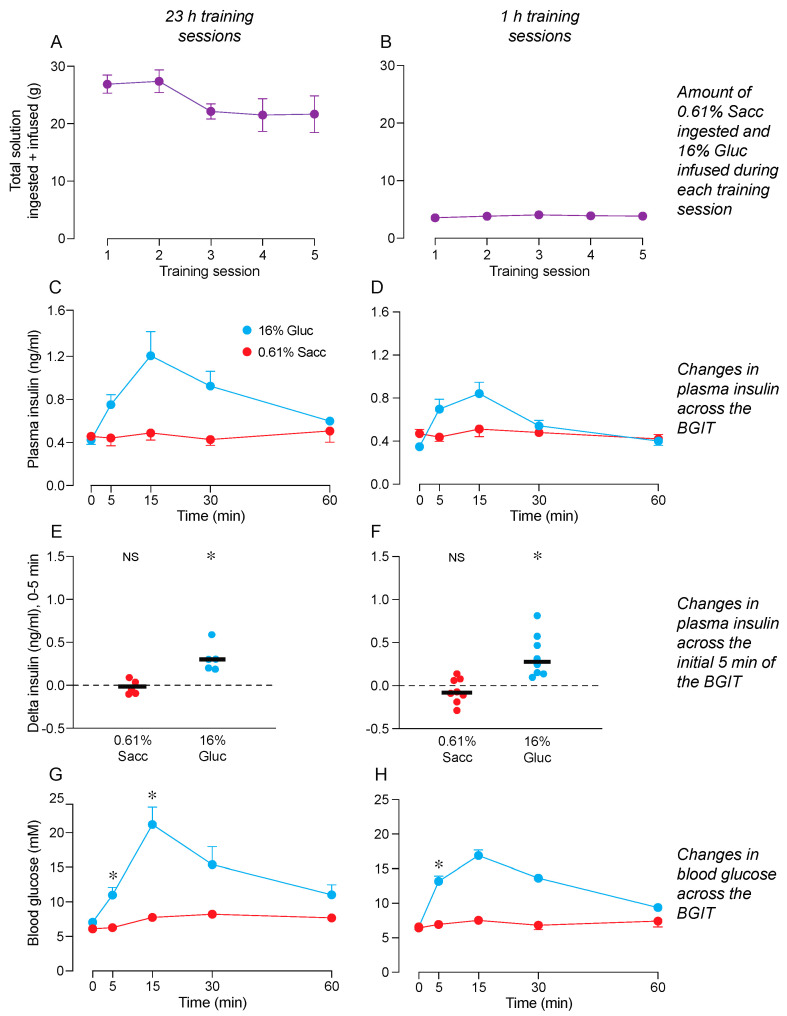
Pairing consumption of 0.61% Sacc with intragastric (IG) co-infusions of 16% Gluc failed to condition a CPIR to 0.61% Sacc (Experiment 1). We show the total amount of solution that was both ingested orally + infused IG, separately for the 23 h (**A**) and 1 h (**B**) training sessions. Subsequently, the mice were subjected to two 60 min BGITs. In the first, the mice licked 0.61% Sacc; in the second, the mice licked for 16% Gluc (a positive control stimulus). We show time-dependent changes in plasma insulin across the entire BGIT (**C**,**D**). We also show insulin responses during the initial 5 min of the BGIT (**E**,**F**). We also show changes in blood glucose across the entire BGIT (**G**,**H**). We asked whether the IG infusions of 16% Gluc during the training sessions caused 0.61% Sacc to elicit a systematic change in plasma insulin, relative to baseline, during the initial 5 min of the BGIT, using a one-sample *t*-test (NS, *p* ≥ 0.05, * *p* < 0.05). Finally, we asked whether blood glucose levels for mice in each treatment increased above baseline during the initial 5 min of the BGIT, using a paired *t*-test (* *p* < 0.05). See Appendix A for further analyses of the data in this figure. In most panels, we show mean ± S.E. In panels (**E**,**F**), we indicate mouse’s score with a circle, and the mean response of each treatment group with a horizonal line. *n* = 5 for mice subjected to 23 h training sessions, and *n* = 8 for mice subjected to 1 h training sessions.

**Figure 2 nutrients-16-02250-f002:**
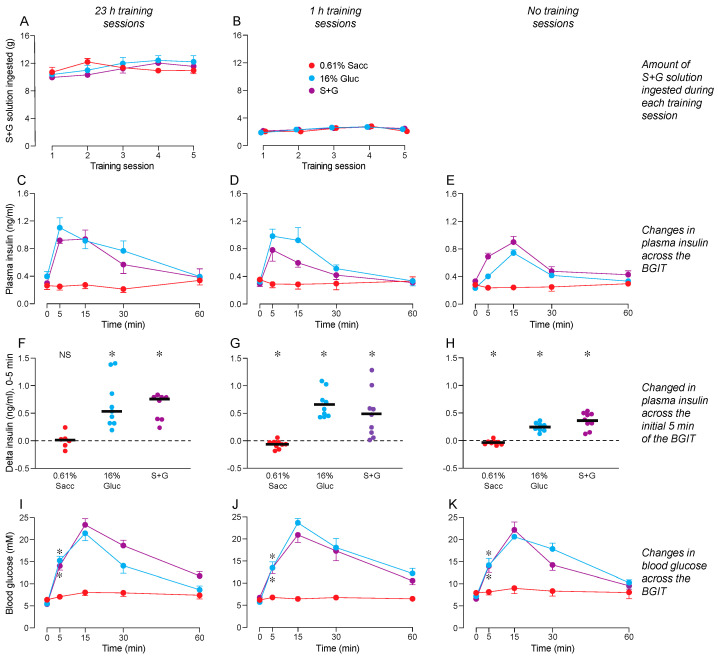
Consumption of the 0.61% Sacc + 16% Gluc (S + G) mixture failed to condition a CPIR to 0.61% Sacc (Experiment 2). We show the amount of S + G consumed by mice during the 23 h (**A**) and 1 h (**B**) training sessions. Within each panel, we distinguish intake by mice subjected to a 60 min BGIT with Sacc, Gluc, or S + G. We show time-dependent changes in plasma insulin across the 60 min BGIT (**C**–**E**) or the initial 5 min of the BGITs (**F**–**H**) in mice with 23 h, 1 h, or no training with the S + G solution. We also show changes in blood glucose across the BGIT (**I**–**K**). In (**F**–**H**), we asked whether the training sessions caused the 0.61% Sacc, 16% Gluc, or S + G solution to elicit a systematic change in plasma insulin, relative to baseline, during the initial 5 min of the BGIT, using one-sample *t*-tests (NS, *p* ≥ 0.05, * *p* < 0.05). In (**I**–**K**), we asked whether blood glucose levels for mice increased above baseline during the initial 5 min of the BGIT in (**I**–**K**), using paired *t*-tests (* *p* < 0.05). See Appendix A for further statistical analyses of the data in this figure. In most panels, we show mean ± S.E. In panels (**F**–**H**), we indicate each mouse’s score with a circle, and the mean response per treatment group with a horizonal line. *n* = 6–9 mice/treatment group.

**Figure 3 nutrients-16-02250-f003:**
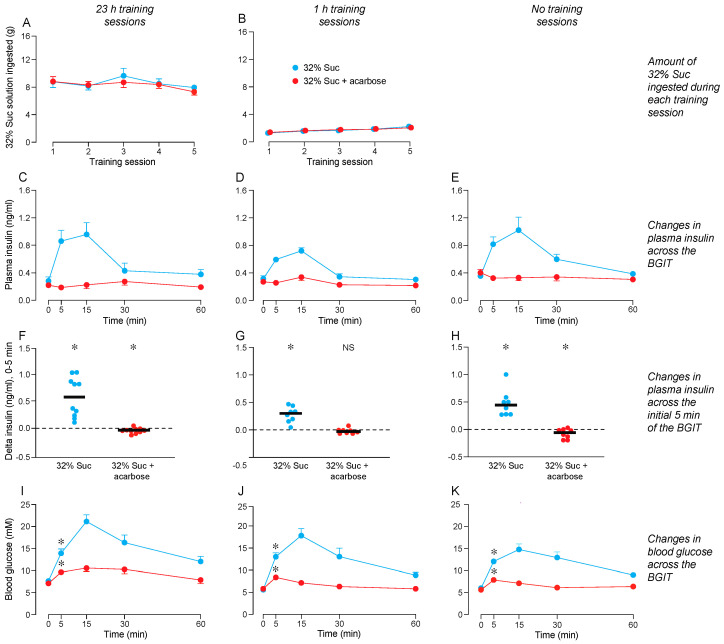
Neither the 1 h nor 23 h training sessions with 32% Suc conditioned a CPIR to 32% Suc + acarbose (Experiment 3). We show the amount of 32% Suc consumed by mice during the 23 h (**A**) and 1 h (**B**) training sessions. Within each panel, we distinguish by mice subjected to a 60 min BGIT with either 32% Suc or 32% Suc + acarbose. We present time-dependent changes in plasma insulin across the entire 60 min BGIT (**C**–**E**) or the initial 5 min of the BGIT (**F**–**H**) in mice with 23 h, 1 h, or no training sessions. We also show changes in blood glucose across the entire BGIT (**I**–**K**). In (**F**–**H**), we asked whether the training sessions caused 32% Suc + acarbose or 32% Suc to elicit a systematic change in plasma insulin, relative to baseline, during the initial 5 min of the BGIT, using one-sample *t*-tests (* *p* < 0.05). In (**I**–**K**), we asked whether blood glucose levels for mice in each treatment group increased above baseline during the initial 5 min of the BGIT, using paired *t*-tests (* *p* < 0.05). See Appendix A for further statistical analyses of these data. In most panels, we show mean ± S.E. In panels (**F**–**H**), we indicate each mouse’s score with a circle, and the mean response per treatment group with a horizonal line. *n* = 8–10 mice per treatment group.

**Figure 4 nutrients-16-02250-f004:**
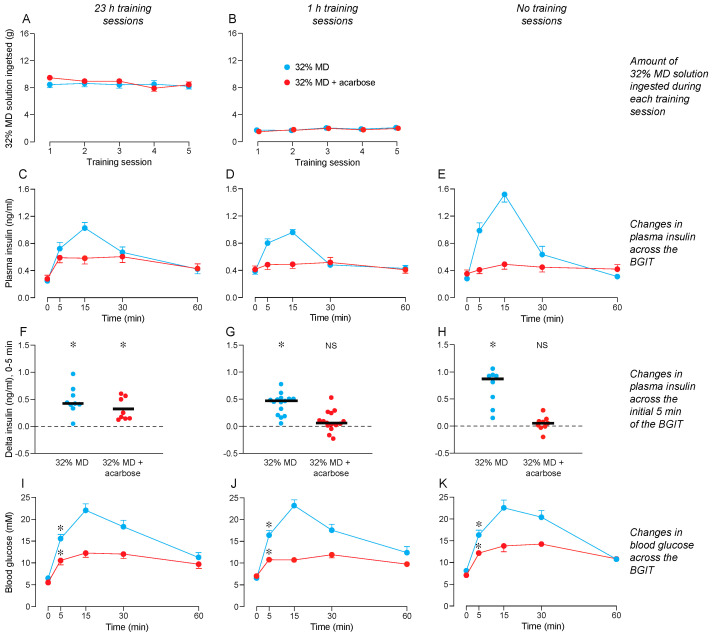
The 23 h training sessions with 32% MD conditioned a CPIR to 32% MD + acarbose (Experiment 4). We show the amount of 32% MD consumed by mice during the 23 h (**A**) and 1 h (**B**) training sessions. Within panels (**A**) and (**B**), we distinguish intake by mice that were later subjected to a 60 min BGIT with either 32% MD + acarbose or 32% MD. We present time-dependent changes in plasma insulin across the entire BGIT (**C**–**E**) or the initial 5 min of the BGIT (**F**–**H**) in mice that were subjected to 23 h, 1 h, or no training sessions. We also show changes in blood glucose across the entire BGIT (**I**–**K**). In (**F**–**H**), we asked whether the training sessions caused 32% MD + acarbose or 32% MD to elicit a systematic change in plasma insulin, relative to baseline, during the initial 5 min of the BGIT, using one-sample *t*-tests (NS, *p* ≥ 0.05, * *p* < 0.05). In (**I**–**K**), we asked whether blood glucose levels for mice in each treatment group increased above baseline during the initial 5 min of the BGIT, using paired *t*-tests (* *p* < 0.05). See Appendix A for further statistical analyses of these data. In most panels, we show mean ± S.E. In panels (**F**–**H**), we represent each mouse’s score with a circle, and the mean response per treatment group with a horizonal line. *n* = 8–10 mice per treatment group.

**Figure 5 nutrients-16-02250-f005:**
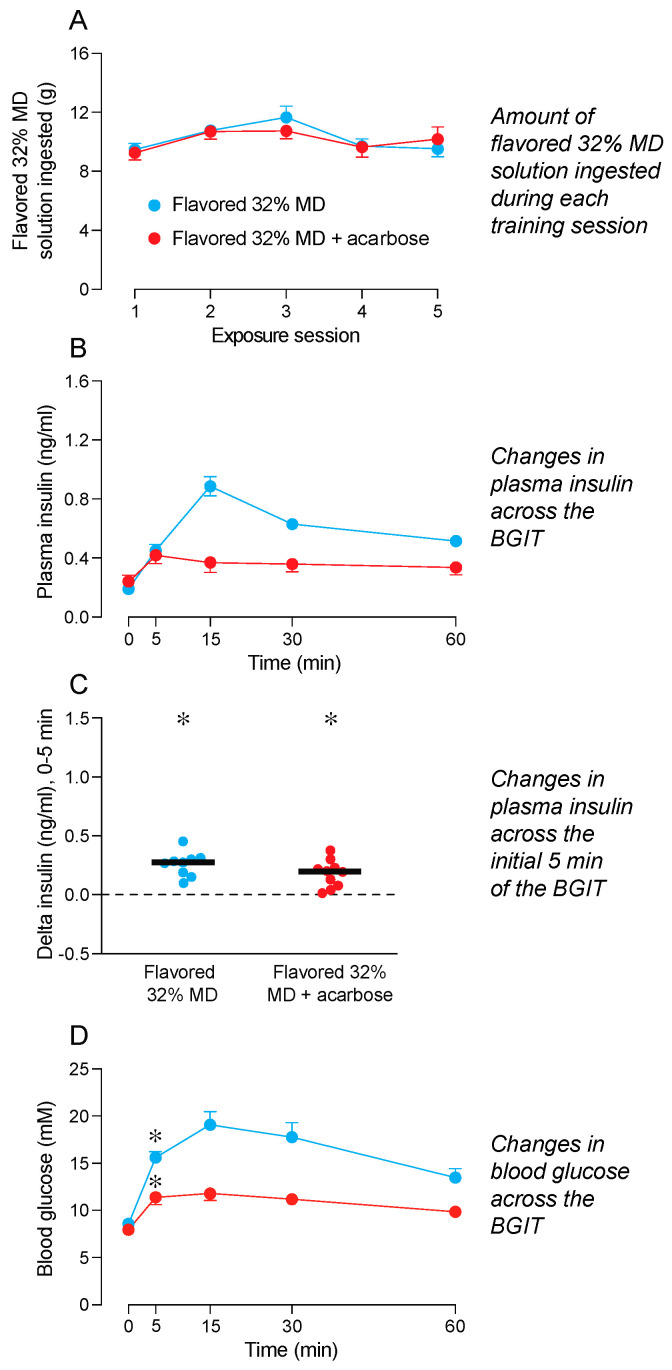
The 23 h training sessions with flavored 32% MD conditioned a CPIR to flavored 32% MD + acarbose (Experiment 5). We show the amount of flavored 32% MD that was consumed during the 23 h training sessions (**A**). We distinguish intake by mice that were later subjected to a 60 min BGIT with either 32% MD + acarbose or 32% MD. We illustrate time-dependent changes in plasma insulin across the entire BGIT (**B**) or the initial 5 min of the BGIT (**C**). We also show changes in blood glucose across the entire BGIT (**D**). In (**C**), we asked whether the training sessions caused the flavored 32% MD or the flavored 32% MD + acarbose to elicit a systematic change in plasma insulin, relative to baseline, during the initial 5 min of the BGIT, using one-sample *t*-tests (* *p* < 0.05). In (**D**), we also asked whether blood glucose levels for mice in each treatment group increased above baseline during the initial 5 min of the BGIT, using paired *t*-tests (* *p* < 0.05). See Appendix A for further statistical analyses of these data. In most panels, we show mean ± S.E. In panel (**C**), we indicate each mouse’s score with a circle, and the mean response of each treatment group with a horizonal line. *n* = 10–11 mice per treatment group.

**Figure 6 nutrients-16-02250-f006:**
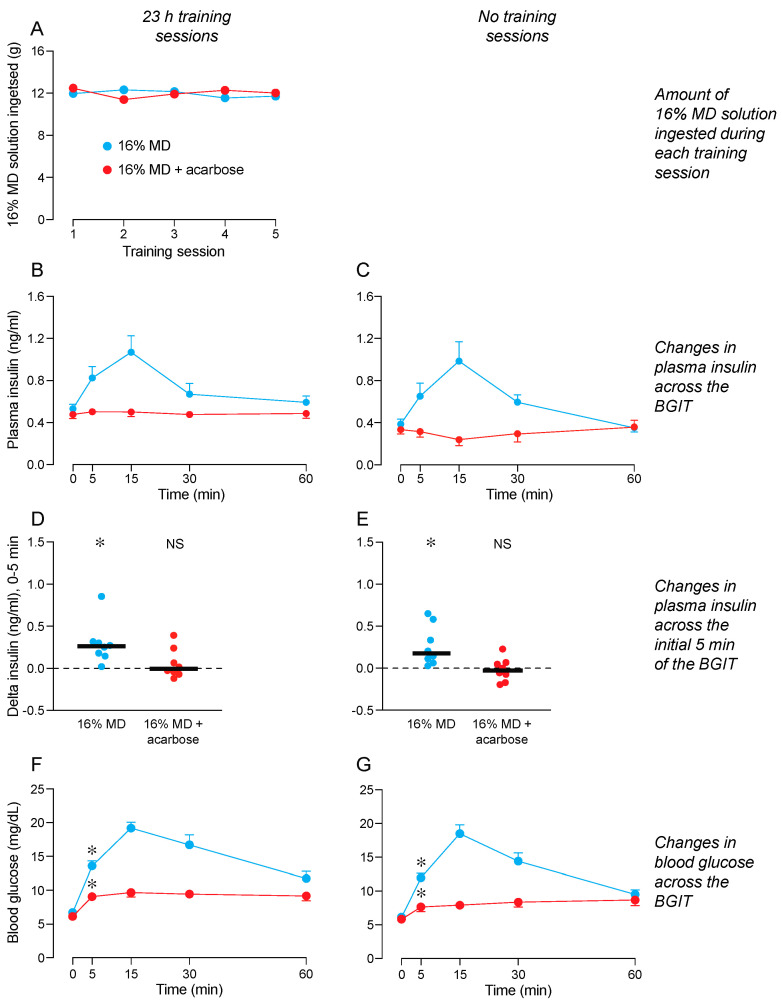
The 23 h training sessions with 16% MD failed to condition a CPIR to 16% MD + acarbose (Experiment 6). We show the amount of 16% MD solution consumed during the 23 h training sessions (**A**). We distinguish intake by mice that were later subjected to a 60 min BGIT with either 16% MD + acarbose or 16% MD. We present time-dependent changes in plasma insulin across the entire BGIT (**B**,**C**) or the initial 5 min of the BGIT (**D**,**E**) in mice with 23 h, 1 h, or no training with the S + G solution. We also show changes in blood glucose across the entire BGIT (**F**,**G**). In (**D**,**E**), we asked whether the training sessions caused 16% MD + acarbose or 16% MD to elicit a systematic change in plasma insulin, relative to baseline, during the initial 5 min of the BGIT, using one-sample *t*-tests (NS, *p* ≥ 0.05, * *p* < 0.05). In (**F**,**G**), we asked whether blood glucose levels for mice in each treatment group increased above baseline during the initial 5 min of the BGIT, using paired *t*-tests (* *p* < 0.05). See Appendix A for further statistical analyses of these data. In most panels, we show mean ± S.E. In panels (**D**,**E**), we represent each mouse’s score with a circle, and the mean response per treatment group with a horizonal line. *n* = 8 mice per treatment group.

**Figure 7 nutrients-16-02250-f007:**
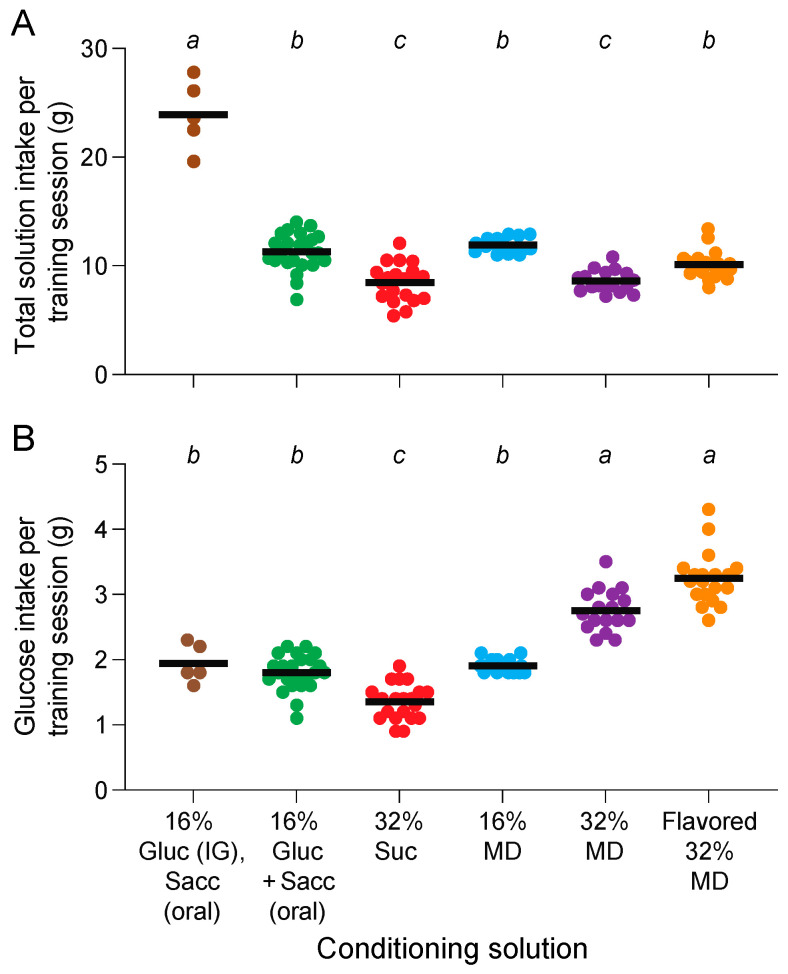
Daily intake of the test solutions during the 23 h training sessions in Experiments 1–6 (**A**). We also show estimated amounts of glucose obtained from each of the test solutions (following digestion) during the same training sessions (**B**). We represent each mouse’s intake with a circle, and the mean intake for each carbohydrate solution with a horizonal line. Within each panel, we indicate the means that differ from one another with unique letters (a, b, c), according to Tukey’s multiple comparison test (*p* < 0.05).

**Figure 8 nutrients-16-02250-f008:**
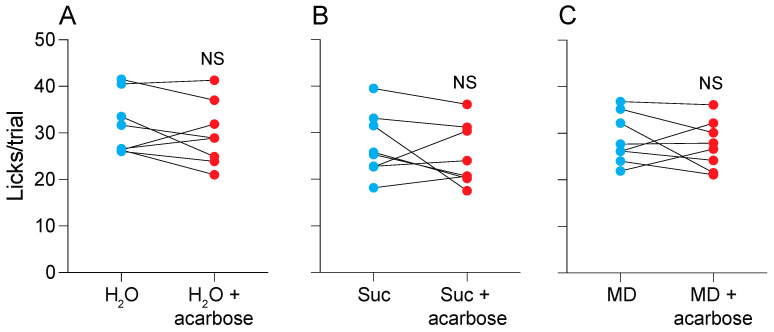
Acarbose (5 mM) did not alter licking responses of mice to water (H_2_O), 32% Suc, or 32% MD. We show the impact of acarbose on the acceptability of (**A**) water, (**B**) 32% Suc, and (**C**) 32% MD. Acceptability was determined by measuring lick rates (i.e., number of licks per 5 s trial) across the two-bottle acceptability tests. We compare lick rates for H_2_O vs. H_2_O + acarbose, 32% Suc vs. 32% Suc + acarbose, and MD vs. MD + acarbose. During the lick test, the mouse was provided access to each of the solutions during separate 5 s trials. The order of presentation of the two solutions across successive trials was determined by a randomized block design. We represent lick rates for each mouse, with two points connected by a line. We used different mice in each panel (*n* = 8/panel). To motivate licking, the mice in panel (**A**) were water-deprived 23 h prior to testing; and the mice in panels (**B**,**C**) were food- and water-restricted 23 h prior to testing. In each panel, there was no difference in lick rate for the two solutions (NS, *p* > 0.05, according to paired *t*-tests).

## Data Availability

The data supporting the findings of this study are available from the corresponding author, J.I.G., upon reasonable request.

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
