# Peer review of "Mice Condition Cephalic-Phase Insulin Release to Flavors Associated with Postoral Actions of Concentrated Glucose"

_nutrients, 2024, doi:10.3390/nu16142250_

Round 1

Reviewer 1 Report

Comments and Suggestions for Authors

The paper by Glendinning and colleague’s provides high detail on experiments to investigate the oral and post-oral stimulus required to condition a cephalic-phase insulin response (CPIR) in C57BL/6 mice, and the context of multiple prior 1- or 23-hour training on these outcomes. The authors show that a high post-post oral glucose stimulus, 32% maltodextrin here, was required to elicit CIPR, adding valuable knowledge on differences between CIPR conditioning and flavor preference in mice, as well as the pathway by which mice may respond to the availability of such a rare high carbohydrate diet.

The paper is well written and study weaknesses are generally well considered, as is the inclusion of positive and negative control conditions, a study strength. A potential concern that needs to be addressed is the use of the moderate+ stressor of multiple 23-hour water deprivation cycles in mice, and high concentrations of tested hexoses. These require further explanation to appreciate the physiological nature of the CIPR the authors contend.

Specific Points

How were potential differences in C57BL/6 source compared in 1-hr tests?

The multiple, close rounds of 23-hour water deprivation (moderate-severe stress) in mice here raise questions on findings based on the “physiological” nature of CPIR measured. The authors should address this potential limit to interpretation.

An intervention figure would improve clarity on methods used substantially.

 It is unclear why groups as low as N = 5 (experiment 1) and disparate as N = 11 were used. The authors should outline and defend this rationale within the manuscript. The a priori basis of this CIPR power calculations should be referenced, not the G*Power software.

Fig 1 - 6 PDF text conversion error

This reviewer agrees, minor saccharin effects to lower plasma insulin during BGIT seem a power-statistical finding, rather than biological

Line 597 space issue.

Expect discussion to begin with an executive summary of findings in context before individual results are addressed.

Author Response

Comments and Suggestions for Authors

The paper by Glendinning and colleague’s provides high detail on experiments to investigate the oral and post-oral stimulus required to condition a cephalic-phase insulin response (CPIR) in C57BL/6 mice, and the context of multiple prior 1- or 23-hour training on these outcomes. The authors show that a high post-post oral glucose stimulus, 32% maltodextrin here, was required to elicit CIPR, adding valuable knowledge on differences between CIPR conditioning and flavor preference in mice, as well as the pathway by which mice may respond to the availability of such a rare high carbohydrate diet.

The paper is well written and study weaknesses are generally well considered, as is the inclusion of positive and negative control conditions, a study strength. A potential concern that needs to be addressed is the use of the moderate+ stressor of multiple 23-hour water deprivation cycles in mice, and high concentrations of tested hexoses. These require further explanation to appreciate the physiological nature of the CIPR the authors contend.

We have added text to address the reviewer’s concerns about the use of water-deprivation and high concentrations of hexoses.

 Specific Points

How were potential differences in C57BL/6 source compared in 1-hr tests?

We do not think that this was an issue, given that the CPIR magnitudes in response to Sacc and Gluc were virtually identical in Figs. 1E (B6 mice from Jackson Lab) and 1F (B6 mice from Envigo).

The multiple, close rounds of 23-hour water deprivation (moderate-severe stress) in mice here raise questions on findings based on the “physiological” nature of CPIR measured. The authors should address this potential limit to interpretation.

We agree that water deprivation causes moderate stress. For this reason, we limited its use during Experiments 1-6 to the day immediately preceding each BGIT. Accordingly, the mice experienced water deprivation twice during Exp 1, and once during Exp 2-6. Further, please note that during Exp 1, we interposed a recovery day between the two water deprivation periods. It follows that the mice did not receive “multiple, close rounds of 23-hour water deprivation.”

An intervention figure would improve clarity on methods used substantially

We agree. In the revision, we have an intervention figure for each experiment (see Suppl. Figs 1-6).

It is unclear why groups as low as N = 5 (experiment 1) and disparate as N = 11 were used. The authors should outline and defend this rationale within the manuscript. The a priori basis of this CIPR power calculations should be referenced, not the G*Power software.

As we note in the revision, we based our power analyses on pilot studies. Because the standard deviation of the CPIR estimates (collected during the pilot studies) varied across the sapid solutions, the power analysis recommended different numbers of mice for each experiment. In response to the reviewer’s comment, we provided some of the input parameters for the power analyses that we ran.

Fig 1 - 6 PDF text conversion error

We think that we have resolved this problem.  

This reviewer agrees, minor saccharin effects to lower plasma insulin during BGIT seem a power-statistical finding, rather than biological.

We are glad that the reviewer concurs with our interpretation of this finding.

Line 597 space issue.

We assume that this space issue will be resolved by the publisher during the final editing process.

Expect discussion to begin with an executive summary of findings in context before individual results are addressed.

In the revision, we have provided a more complete executive summary of findings in the Conclusion section. To limit redundancy, we have avoided providing a second executive summary at the beginning of the Discussion section.

Reviewer 2 Report

Comments and Suggestions for Authors

The paper “Mice condition cephalic-phase insulin release to flavors associated with postoral actions of concentrated glucose” asked whether mice can condition a CPIR to the flavor of a sapid solutions that produce postoral glucose stimulation. The article is very interesting, but there are some questions that need to be further improved or explained.

Comments:

Q1. In the introduction, it is suggested to supplement or strengthen the research significance, or the application scenarios of the results of this paper.

Q2. The description of the experimental method is lengthy, and it is suggested to draw a schematic diagram of the experimental process to further reduce the difficulty of readers' understanding.

Q3. Garbled code appears in Figure 1, please check, as well as other Figures.

Q4. The author describes the induction of insulin by various sugars. Why then is the response to insulin by sucrose and glucose so different? Does the result of this study mean that consuming sucrose (or similar sugars that do not cause an increase in insulin levels) is less likely to result in diabetes than consuming glucose? Does it also mean that in cases of low blood sugar, supplementing with sucrose (or similar sugars that do not cause an increase in insulin levels) will have poor effects?

Q5. Did the results of this study merely arise from the influence of different constituents' taste? Or was there also involvement of the organism's polysaccharide enzymes or other metabolic signaling pathways? Additionally, the results and discussion sections of this paper are somewhat scattered, and a more effective summary would be beneficial, such as using a diagram to illustrate the points, which would greatly enhance the presentation.

Q6. The conclusion should be further refined. Furthermore, the conclusion section should not contain any references and ought to exclusively encapsulate the authors' synthesis of the study's results.

Author Response

Comments and Suggestions for Authors

The paper “Mice condition cephalic-phase insulin release to flavors associated with postoral actions of concentrated glucose” asked whether mice can condition a CPIR to the flavor of a sapid solutions that produce postoral glucose stimulation. The article is very interesting, but there are some questions that need to be further improved or explained.

Comments:

Q1. In the introduction, it is suggested to supplement or strengthen the research significance, or the application scenarios of the results of this paper.

We have added text to the Introduction to strengthen the research significance.

Q2. The description of the experimental method is lengthy, and it is suggested to draw a schematic diagram of the experimental process to further reduce the difficulty of readers' understanding.

We agree that the Methods section is lengthy. This stems from large number of experiments we performed. We have made every effort to make this section as succinct as possible.

To help clarify the nature of experimental interventions, we have provided graphical timelines for each of the conditioning experiments (see Suppl. Figs 1-6).

Q3. Garbled code appears in Figure 1, please check, as well as other Figures.

We think that we have resolved this problem.

Q4. The author describes the induction of insulin by various sugars. Why then is the response to insulin by sucrose and glucose so different? Does the result of this study mean that consuming sucrose (or similar sugars that do not cause an increase in insulin levels) is less likely to result in diabetes than consuming glucose? Does it also mean that in cases of low blood sugar, supplementing with sucrose (or similar sugars that do not cause an increase in insulin levels) will have poor effects?

Unfortunately, our study does not directly address any of these interesting questions. Our goal was to determine whether mice could condition CPIR to the flavor of sapid solutions that cause postoral glucose stimulation. We selected test solutions that did not elicit CPIR in naïve mice. Our results indicate that the level of postoral glucose stimulation is paramount in determining whether mice will condition a CPIR to the flavor of a cephalic stimulus.

Q5. Did the results of this study merely arise from the influence of different constituents' taste? Or was there also involvement of the organism's polysaccharide enzymes or other metabolic signaling pathways?

We thank the reviewer for encouraging us to clarify these important issues. As we note in the revision, we think that the intensity of postoral glucose stimulation was more important than flavor quality of the training stimulus in determining whether CPIR conditioning occurred.

Additionally, the results and discussion sections of this paper are somewhat scattered, and a more effective summary would be beneficial, such as using a diagram to illustrate the points, which would greatly enhance the presentation.

We agree that the use of many different experiments has made the paper complicated to read and interpret. We have done our best to address this structural problem.

While we were unable to develop a diagram to integrate the many different elements of our study, we tried to accomplish this integration in the revised Conclusion.

Q6. The conclusion should be further refined. Furthermore, the conclusion section should not contain any references and ought to exclusively encapsulate the authors' synthesis of the study's results.

We have modified the conclusions to address these concerns. We have removed most of the references, but left a few as they were critical to support our comparison with flavor-conditioning studies.